# Morphologic changes of the no-touch saphenous vein as Y-composite versus aortocoronary grafts (CONFIG Trial)

Suk Ho Sohn[1], Yoonjin Kang[1], Ji Seong Kim[1], Jeehoon Kang[2], Ho Young Hwang[1]*

1 Department of Thoracic and Cardiovascular Surgery, Seoul National University Hospital, Seoul National University College of Medicine, Seoul, Republic of Korea, 2 Department of Internal Medicine and Cardiovascular Center, Seoul National University Hospital, Seoul National University College of Medicine, Seoul, Republic of Korea

☯ These authors contributed equally to this work.
* scalpel@snu.ac.kr

## Abstract

### Objective

This randomized controlled trial was aimed to compare 1-year morphologic changes of the no-touch saphenous vein graft as Y-composite (Composite group) versus aortocoronary (Aorta group) configurations in coronary artery bypass grafting.

### Methods

The primary endpoint was intima-media thickness of the saphenous vein graft as measured by intravascular ultrasound (IVUS) at the 1-year angiographic evaluation. Recruitment of 25 patients in each group was necessary based on a superiority design. Among the 50 patients, IVUS data were obtained in 22 and 24 patients from the Composite and Aorta groups, respectively.

### Results

Mean age was 64.8 ± 9.2 years, and the proportion of females was 20.0%. The numbers of distal anastomoses per saphenous vein graft were 2.7 ± 1.1 and 2.6 ± 0.8 in the Composite and Aorta groups, respectively. The intima-media thickness of the saphenous vein graft 1 year after surgery were 0.25 ± 0.04 mm and 0.24 ± 0.06 mm in the Composite and Aorta groups, respectively (*P for superiority* = .99). Other IVUS parameters of saphenous vein grafts, including vessel diameter, luminal diameter, and the ratio of intima-media thickness to vessel diameter, also demonstrated no differences between the groups. No neointimal hyperplasia or plaque formation was detected using IVUS. All study patients underwent 1-year angiographic evaluation, and the patency rates were 94.7%(89 out of 94 anastomoses) and 100.0%(90 out of 90 anastomoses) in the Composite and Aorta groups, respectively.

**Data availability statement:** All relevant data are within the manuscript and its Supporting Information files.

**Funding:** The author(s) received no specific funding for this work.

**Competing interests:** The authors have declared that no competing interests exist.

**Abbreviations:** CABG = coronary artery bypass grafting, IQR = interquartile range, ITA = internal thoracic artery, IVUS = intravascular ultrasound, LAD = left anterior descending artery, LCX = left circumflex artery, LITA = left internal thoracic artery, NT = no-touch, OPCAB = off-pump coronary artery bypass grafting, PVAT = perivascular adipose tissue, QCA = quantitative coronary angiography, RCA = right coronary artery, RM ANOVA = repeated measures analysis of variance, SV = saphenous vein.

## Conclusions

The intima-media thickness of the saphenous vein graft 1 year after surgery demonstrated no significant difference between the Y-composite and aortocoronary configurations (NCT04782492).

**Trial Registration:** ClinicalTrials.gov NCT04782492.

## Introduction

Current guidelines recommend multiarterial grafting (MAG) during coronary artery bypass grafting (CABG) based on several randomized trials and matched analyses [1,2]. However, single arterial grafting using saphenous vein (SV) grafts is used worldwide, and SV grafts are the most commonly used second conduit of choice in CABG [3,4]. The SV has many advantages as a bypass conduit such as ease of handling, availability of graft material, customizable length, less demanding harvesting techniques, and resistance to spasm. Therefore, efforts have been made to obtain optimal graft quality of the SV and improve its long-term patency.

The novel no-touch (NT) technique of SV harvesting, in which the SV is harvested with its surrounding perivascular tissue and treated with minimal manipulation, and manual intraluminal distension of the SV is avoided, was first introduced in 1996 [5]. The superior patency of the SV using the NT technique (NT-SV) compared to conventional harvesting has been shown in many studies, which led to a IIa recommendation in recent guidelines [2]. The NT technique demonstrated preservation of endothelial cell integrity, expression of endothelial nitric oxide (NO) synthase, and reduced vasoconstriction compared to conventional techniques [6].

In addition to the harvesting technique, previous studies suggested that the use of the NT-SV as a composite graft based on the in situ left internal thoracic artery (L[ITA]) may be beneficial for graft patency up to 10 years after surgery [7,8]. The patency of the NT-SV composite grafts was noninferior to the right ITA composite grafts in these studies, with 10-year patency rates of 93.1% and 96.6%, respectively. Although possible explanations for the use of SV as a composite graft have been suggested, whether this favorable patency of NT-SV composite grafts is due to the effect of the composite grafting strategy or the effect of NT harvesting is not clear.

Therefore, the present randomized controlled trial, entitled as 'Morphologic **C**hanges **O**f the **N**o-touch saphenous vein as Y-composite versus aortocoronary gra**F**ts **I**n coronary artery bypass **G**rafting (CONFIG)', compared the 1-year morphologic features of the NT-SV used as Y-composite grafts and those used as aortocoronary grafts (ClinicalTrials.gov identifier: NCT04782492).

## Materials and methods

### Study design

The institutional review board approved the study protocol (approval date 03/10/2021, approval no. H-2101-143-1191), and written informed consents were obtained from

all study patients. The study was designed according to the Consolidated Standards of Reporting Trials statement [9]. Patients over 19 years of age who were scheduled to undergo primary isolated CABG for multivessel disease on a none-mergency basis and for whom the use of the LITA and SV as bypass grafts was planned were assessed for eligibility for study enrollment. The exclusion criteria included (1) patients undergoing concomitant cardiac procedures, including valve or aorta surgery; (2) patients in whom it was not feasible to use the LITA or SV as a bypass graft due to intrinsic problems of the conduits; (3) patients with severely atherosclerotic or calcified ascending aorta that precluded the use of aortocoronary anastomosis of the SV; (4) patients who had vasculitis; (5) patients with severe comorbidities that might limit the performance of a 1-year angiographic evaluation; and (6) patients who refused to participate in the study.

Between July 8th 2021 and October 25th 2022, 85 patients were assessed for eligibility, and 25 patients were excluded due to poor SV quality (n = 9), poor LITA quality (n = 4), ascending aorta disease (n = 6), critical preoperative status (n = 4) or vasculitis (n = 2). Of the 60 patients who were eligible for the study, 10 patients refused to participate, and 50 patients were enrolled. The study patients were randomly assigned to the Y-composite group (the Composite group) or aortocoronary group (the Aorta group) in a 1:1 manner (Fig 1).

## Operative strategies and randomization process

Off-pump CABG (OPCAB) was attempted in the study patients regardless of left ventricular function. The in situ LITA was harvested using a skeletonization technique. The SV was simultaneously harvested, preferentially from the lower leg, using an NT technique that retained perivascular soft tissue, as previously described [10]. Randomization was performed after the conduits were harvested without injury, and after the intraoperative findings confirmed that either of the grafting strategies was feasible. Web-based block randomization was performed using randomly determined block sizes of 4 and 6. After randomization, the

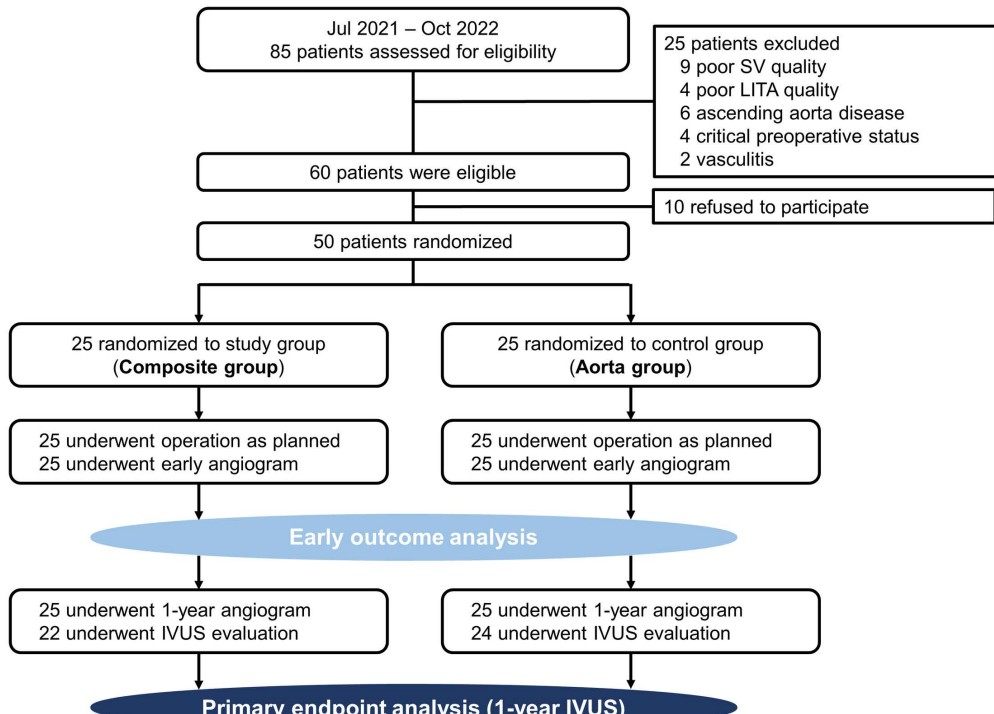

**Fig 1. Flow diagram of the study population.**

procedures were performed according to the assigned grafting strategy. Proximal anastomosis of the NT-SV to the in situ LITA or ascending aorta was performed first. A proximal sealing system (the Heartstring III Proximal Seal System [MAQUET Holding B.V. & Co. KG, Rastatt, Germany]) was routinely used in the Aorta group. While the SV was left to be dilated by native pressure of the LITA or aorta, distal anastomosis of the LITA to the left anterior descending artery (LAD) territory was performed first. The SV was then anastomosed to the diagonal branch, to the vessels in the left circumflex coronary artery (LCX) territory, and to the vessels in the right coronary artery (RCA) territory as needed using a sequential anastomotic technique (Fig 2).

## Histologic evaluation of the saphenous vein

Immediately after harvesting, the proximal anastomotic end of the SV, approximately 2 mm in length, was sampled and preserved as formalin-fixed paraffin-embedded tissue for histologic evaluation. The histologic evaluation compared the intima-media thickness of the SV between the groups as a baseline reference and evaluated the quality of the SV. The specimen was divided into 3 cross-sections. Histologic evaluation was performed using hematoxylin and eosin staining and Masson's trichrome staining. The intima-media thickness was measured in 4 directions for each section, and the values obtained for the 3 sections were averaged (S1 Fig).

## Angiographic evaluation of graft patency

Graft angiograms were obtained 1 day and 1 year after surgery. Graft failure was defined when the grafts were either completely occluded or diffusely narrowed (the string phenomenon) [11]. Competitive flow of a bypass graft was defined

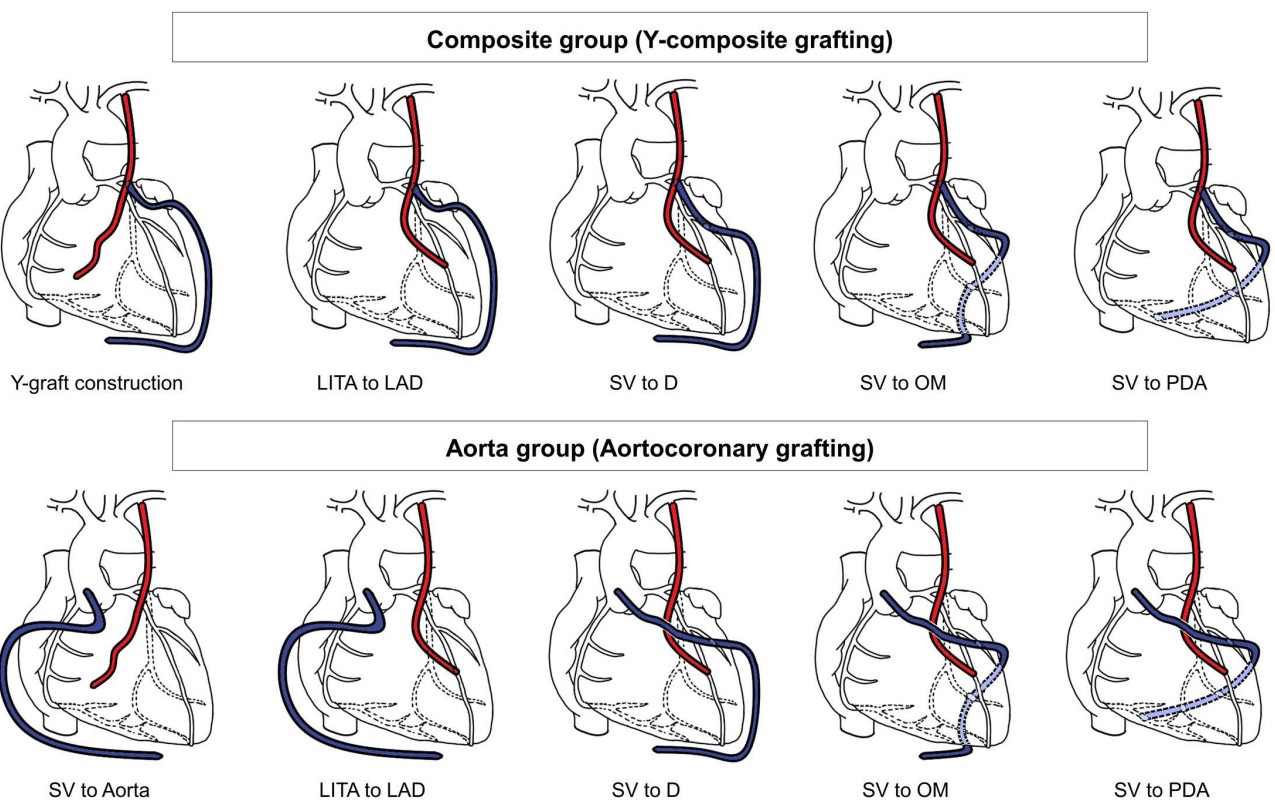

**Fig 2. Procedural sequences for each grafting strategy. (A)** Y-composite grafting strategy, and **(B)** aortocoronary grafting strategy. D, diagonal branch; LAD, left anterior descending artery; LITA, left internal thoracic artery; OM, obtuse marginal; PDA, posterior descending artery; SV, saphenous vein.

when the flow from the bypass graft to the target vessel was not visualized on graft angiography and the graft was filled by retrograde flow from the native target vessel on native coronary angiography [12].

A single experienced cardiologist performed quantitative coronary angiography (QCA) analyses of the grafts using a 6-Fr guiding catheter for calibration and an edge detection system (CAAS 5.7 QCA system, Pie Medical, Maastricht, the Netherlands).

## Intravascular ultrasound evaluation of grafts

Intravascular ultrasound (IVUS) was simultaneously performed during the 1-year angiographic evaluation. IVUS was performed in a standard manner using an automated motorized pullback system (0.5 mm/s) with commercially available imaging catheters (Boston Scientific/SCIMED, Minneapolis, MN). IVUS images were acquired after the administration of 100–200 mg nitroglycerine. For evaluation of the SV graft, the IVUS catheter was advanced and located approximately 5 cm distal to the Y- or aortocoronary anastomosis, and automated motorized pullback was performed. The LITA graft was also evaluated with IVUS: 5 cm proximal and distal to the Y-anastomosis in the Composite group and 10 cm in the mid-LITA in the Aorta group. To avoid the influence of suture materials, areas 1 cm proximal and distal to the Y- or aortocoronary anastomosis were excluded from the analyses (Fig 3). Quantitative analyses of the IVUS data were performed using computerized planimetry software (echoPlaque 3.0, Indec Systems Inc, Santa Clara, CA). The lumen and vessel areas were measured every 2.0 mm. All volumes were calculated using the Simpson rule and normalized for analyzed length. The area of the intima-media was calculated by subtracting the luminal area from the vessel area, and the intima-media thickness was calculated by subtracting the luminal diameter from the vessel diameter (Fig 4).

## Statistical analysis

The primary endpoint of the CONFIG trial was intima-media thickness as measured by IVUS during a 1-year graft angiogram. The secondary endpoints included luminal diameter measured by IVUS and QCA on 1-year angiogram, 1-year

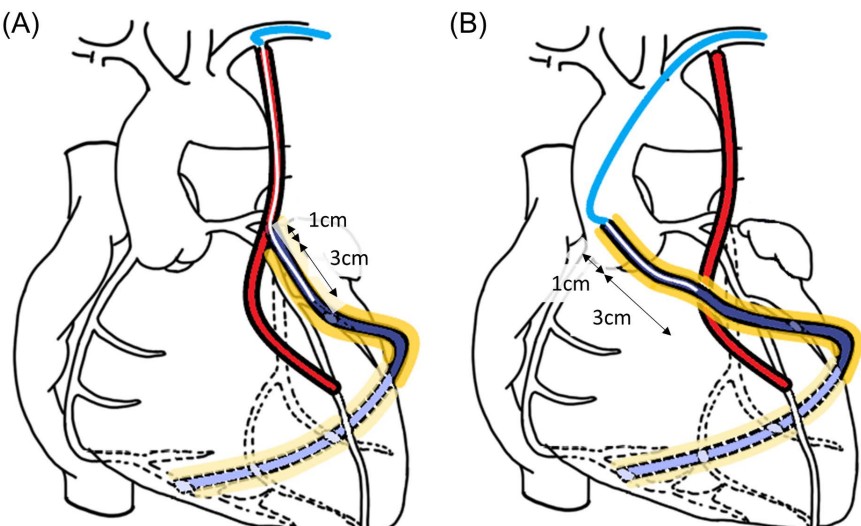

**Fig 3. Schematic diagram of IVUS evaluation for the composite and aorta groups. (A)** In the Composite group, an IVUS catheter was introduced via the orifice of the LITA, and the proximal part of the Y-composite SV graft 3 cm in length was evaluated. SV grafts within 1 cm of the site of Y-anastomosis were excluded from the IVUS evaluation to avoid interference from suture materials. **(B)** In the Aorta group, an IVUS catheter was introduced via the orifice of the aortocoronary anastomosis, and the proximal part of the aortocoronary SV graft 3 cm in length was evaluated. SV grafts within 1 cm of the aortocoronary anastomosis were excluded from the IVUS evaluation to avoid the interference of suture materials.

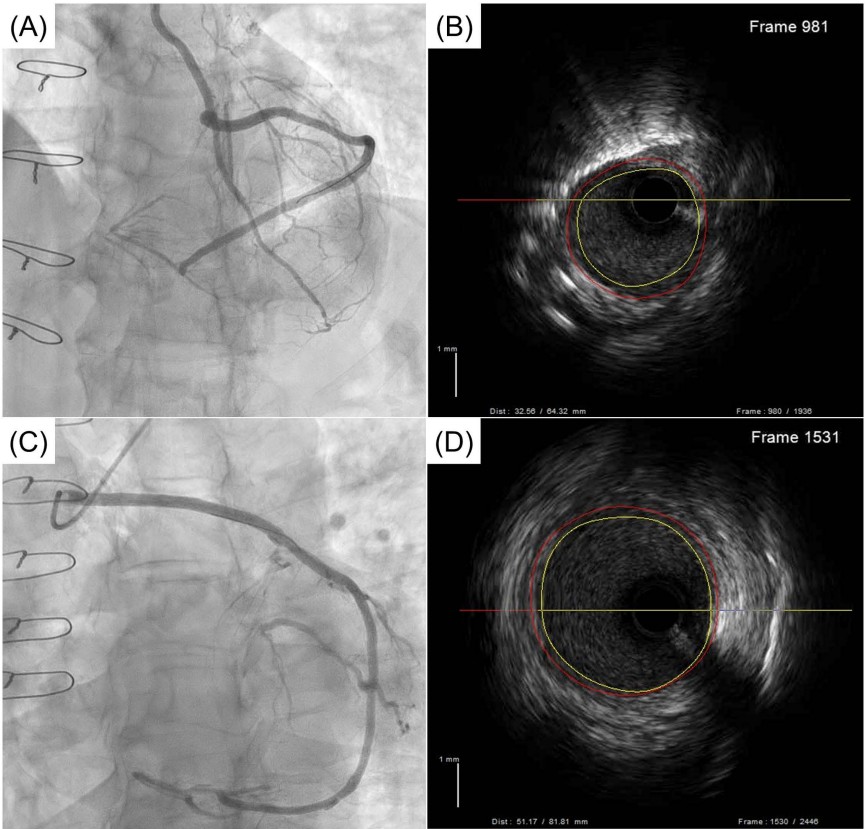

**Fig 4. IVUS evaluation of SV grafts in the composite and aorta groups. (A)** Graft angiogram in the composite group, **(B)** IVUS image in the composite group, (C) graft angiogram in the aorta group, and **(D)** IVUS image in the aorta group. IVUS evaluation demonstrated comparable intima-media thickness between the 2 groups, without any neointimal hyperplasia in either group.

angiographic graft patency, and clinical outcomes, such as all-cause mortality, cardiac death, target vessel revascularization and reintervention at 1 year.

Based on previous studies, the reference intima-media thickness was estimated to be 0.31±0.12 mm for the Y-composite configuration [13] and 0.43±0.09 mm for the aortocoronary configuration [14,15]. With a superiority design, this study was designed to have 90% power to detect a significant difference in intima-media thickness between the 2 groups, with a 2-sided type I error of 5.0%. Based on this power calculation, 17 patients were needed in each group. Allowing for a 30% dropout rate during the 1-year follow-up, the recruitment of 50 patients (25 patients in each group) was necessary.

Statistical analyses were performed using SPSS software (version 29.0; IBM, Armonk, NY) and SAS software (version 9.3; SAS Institute, Cary, NC). Continuous variables are presented as the means±standard deviation for normally distributed variables and as median with interquartile range (IQR) for nonnormally distributed variables. Categorical variables are presented as the number and percentage of subjects. Comparisons between two groups were performed using the chi-square test or Fisher's exact test for categorical variables and Student's t test or the Wilcoxon rank sum test for continuous variables. Comparisons of the serial changes in the luminal diameter of the grafts via QCA analysis were performed using repeated measures analysis of variance (RM ANOVA). For the primary endpoint, multiple imputation was additionally performed to deal with missing data using Fully Conditional Specification (FCS) method. A *P* value of <.05 was considered statistically significant. All outcomes were compared with an intention-to-treat base.

## Results

### Baseline characteristics

The median age of the patients was 65 years (IQR 61, 69), and 20% of the patients (10 out of 50 patients) were female. There were no differences in demographic data or preoperative risk factors between the 2 groups (Table 1).

### Operative data

Forty-nine patients underwent OPCAB as planned. On-pump conversion was needed during off-pump LITA-to-LAD anastomosis in the remaining patient because of hemodynamic instability, and the surgery proceeded as an on-pump beating CABG.

The SV was harvested from the lower leg in 19 (76.0%) and 23 (92.0%) patients in the Composite and Aorta groups, respectively (Table 2). The length of the NT-SVs used were 16.0±4.3 cm and 20.4±3.8 cm in the Composite and Aorta groups, respectively. The average numbers of distal anastomoses per patient were 3.8±1.2 and 3.6±0.8 in the Composite and Aorta groups, respectively ($P$=.58). The average numbers of distal anastomoses per LITA were 1.0±0.2 and 1.0±0.2, respectively, and distal anastomoses per SV were 2.7±1.1 and 2.6±0.8 in the Composite group and Aorta group, respectively ($P$=.57). There were no significant differences between the 2 groups in the average number of distal anastomoses in the LAD, LCX or RCA territories (Table 3).

Table 1. Preoperative characteristics and risk factors of the study patients.

| Variable | Total (n = 50) | Composite group (n = 25) | Aorta group (n = 25) | P |
|---|---|---|---|---|
| Age (years) | 65 (61, 69) | 64 (61, 69) | 65 (61, 69) | .92 |
| Sex (female), n (%) | 10 (20.0) | 4 (16.0) | 6 (24.0) | .48 |
| EuroSCORE II | 0.9 (0.7–1.6) | 0.7 (0.6–2.0) | 1.0 (0.7–1.3) | .41 |
| Risk factors, n (%) | | | | |
| Smoking | 28 (56.0) | 15 (60.0) | 13 (52.0) | .57 |
| Body mass index > 25.0 kg/m$^2$ | 24 (48.0) | 14 (56.0) | 10 (40.0) | .26 |
| Hypertension | 32 (64.0) | 17 (68.0) | 15 (60.0) | .56 |
| Diabetes mellitus | 25 (50.0) | 12 (48.0) | 13 (52.0) | .78 |
| Dyslipidemia | 22 (44.0) | 13 (52.0) | 9 (36.0) | .25 |
| History of stroke | 3 (6.0) | 0 (0.0) | 3 (12.0) | .24 |
| Chronic kidney disease | 10 (20.0) | 8 (32.0) | 2 (8.0) | .03 |
| Chronic obstructive pulmonary disease | 1 (4.0) | 0 (0.0) | 1 (4.0) | >.99 |
| Atrial fibrillation | 0 (0.0) | 0 (0.0) | 0 (0.0) | – |
| Peripheral vascular disease | 9 (18.0) | 3 (12.0) | 6 (24.0) | .46 |
| Previous percutaneous coronary intervention | 13 (26.0) | 6 (24.0) | 7 (28.0) | .75 |
| Left ventricular ejection fraction < 0.35 | 0 (0.0) | 0 (0.0) | 0 (0.0) | – |
| Preoperative diagnosis, n (%) | | | | |
| Stable angina | 26 (52.0) | 12 (48.0) | 14 (56.0) | .57 |
| Acute coronary syndrome | 24 (48.0) | 13 (52.0) | 11 (44.0) | .57 |
| Three vessel disease, n (%) | 35 (70.0) | 17 (68.0) | 18 (72.0) | .76 |
| Left main disease, n (%) | 23 (46.0) | 12 (48.0) | 11 (44.0) | .78 |

Continuous variables were presented as medians with interquartile ranges for non-normally distributed variables.

**Table 2. Location of the harvested saphenous vein.**

|  | Total (n=50) | Composite group (n=25) | Aorta group (n=25) | P |
|---|---|---|---|---|
| Lower leg, n (%) | 42 (84.0%) | 19 (76.0%) | 23 (92.0%) | .25 |
| Left | 31 (62.0%) | 13 (52.0%) | 18 (76.0%) | |
| Right | 11 (22.0%) | 6 (24.0%) | 5 (20.0%) | |
| Upper leg, n (%) | 8 (16.0%) | 6 (24.0%) | 2 (8.0%) | .25 |
| Left | 5 (10.0%) | 5 (20.0%) | 0 (0.0%) | |
| Right | 3 (6.0%) | 1 (4.0%) | 2 (8.0%) | |

**Table 3. Comparison of the numbers of distal anastomoses between the 2 groups.**

| Variables | Total (n=50) | Composite group (n=25) | Aorta group (n=25) | P |
|---|---|---|---|---|
| Per patient | 3.7±1.0 | 3.8±1.2 | 3.6±0.8 | .58 |
| Per LITA | 1.0±0.2 | 1.0±0.2 | 1.0±0.2 | >.99 |
| Per SV | 2.6±1.0 | 2.7±1.1 | 2.6±0.8 | .57 |
| Anastomosed to LAD territory | 0.8±0.5 | 0.8±0.6 | 0.8±0.4 | >.99 |
| Anastomosed to LCX territory | 1.0±0.6 | 1.1±0.7 | 1.0±0.5 | .65 |
| Anastomosed to RCA territory | 0.8±0.6 | 0.8±0.6 | 0.7±0.6 | .66 |

LAD, left anterior descending artery; LCX, left circumflex artery; LITA, left internal thoracic artery; RCA, right coronary artery; SV, saphenous vein.

## Early and 1-year clinical outcomes

There was no operative mortality. Atrial fibrillation (n=11, 22.0%) was the most common complication after surgery. There were no differences in the occurrence rates of postoperative complications between the 2 groups (Table 4).

No patient died during the 1-year follow-up. Reintervention was needed in one patient in the Aorta group due to critical stenosis at the proximal anastomosis site. No other events occurred during the follow-up.

## Early and 1-year angiographic graft patency

All study patients underwent angiographic evaluations at the early postoperative period and 1 year after CABG. Early postoperative graft angiograms were performed on postoperative Day 1 (interquartile range [IQR] 1, 1), and 1-year angiograms were performed 11.5 months (IQR 11.2, 12.2) after CABG. The early postoperative angiographic patency rates were 98.9% (93 out of 94 anastomoses) and 100.0% (90 out of 90 anastomoses) in the Composite and Aorta groups, respectively (P>.99, S1 Table). The 1 occluded anastomosis in the Composite group was an anastomosis that was constructed with the SV to the obtuse marginal artery with a diameter less than 1 mm. Competitive flows were observed at 3 anastomoses only in the Composite group, and all 3 anastomoses were made to the RCA territory with moderate stenosis using SV grafts.

The 1-year angiographic patency rates were 94.7% (89 out of 94 anastomoses) and 100.0% (90 out of 90 anastomoses) in the Composite and Aorta groups (P=.06), respectively (Table 5). Competitive flows were observed at 3 and 1 anastomoses in the Composite and Aorta groups, respectively.

## Data from intravascular ultrasound

One-year IVUS was performed in 22 and 24 patients in the Composite and Aorta groups, respectively. It was not performed on the other 3 patients or on 1 patient due to a shortage of IVUS catheter length needed to reach the target

**Table 4. Comparison of early clinical outcomes between the 2 groups.**

| Variables | Total (n = 50) | Composite group (n = 25) | Aorta group (n = 25) | P |
|---|---|---|---|---|
| Operative mortality, n (%) | 0 (0.0) | 0 (0.0) | 0 (0.0) | – |
| Postoperative complications, n (%) | | | | |
| Postoperative atrial fibrillation | 11 (22.0) | 4 (16.0) | 7 (28.0) | .50 |
| Acute kidney injury | 1 (2.0) | 0 (0.0) | 1 (4.0) | >.99 |
| Respiratory complications | 1 (2.0) | 1 (4.0) | 0 (0.0) | >.99 |
| Delirium | 1 (2.0) | 1 (4.0) | 0 (0.0) | >.99 |
| Bleeding reoperation | 0 (0.0) | 0 (0.0) | 0 (0.0) | – |
| Stroke | 0 (0.0) | 0 (0.0) | 0 (0.0) | – |
| Mediastinitis | 0 (0.0) | 0 (0.0) | 0 (0.0) | – |

**Table 5. Comparison of 1-year angiographic patency rates between the 2 groups.**

| Variables | Total (n = 50) | Composite group (n = 25) | Aorta group (n = 25) | P |
|---|---|---|---|---|
| Overall | 97.3% (179/184) | 94.7% (89/94) | 100.0% (90/90) | .06 |
| LITA | 98.1% (51/52) | 96.2% (25/26) | 100.0% (26/26) | >.99 |
| SV | 97.0% (128/132) | 94.1% (64/68) | 100.0% (64/64) | .12 |
| Anastomosed to LAD territory | 100.0% (42/42) | 100.0% (21/21) | 100.0% (21/21) | – |
| Anastomosed to LCX territory | 98.1% (51/52) | 96.3% (26/27) | 100.0% (25/25) | >.99 |
| Anastomosed to RCA territory | 92.1% (35/38) | 85.0% (17/20) | 100.0% (18/18) | .23 |
| Sequential anastomosis | 100.0% (82/82) | 100.0% (43/43) | 100.0% (39/39) | – |
| Terminal anastomosis | 92.0% (46/50) | 84.0% (21/25) | 100.0% (25/25) | .11 |

LAD, left anterior descending artery; LCX, left circumflex artery; LITA, left internal thoracic artery; RCA, right coronary artery; SV, saphenous vein.

graft. The intima-media thicknesses of the SV grafts, which was the primary endpoint of the trial, were 0.25 ± 0.04 mm and 0.24 ± 0.06 mm in the Composite and Aorta groups, respectively (P = .99). There were no significant differences in other IVUS parameters, including vessel diameter, luminal diameter or intima-media thickness to vessel diameter, between the 2 groups (Table 6). When further analysis was performed using multiple imputation to deal with missing data, the intima-media thicknesses of the SV grafts demonstrated no difference between the groups (0.25 ± 0.06 mm vs. 0.24 ± 0.06 mm in Composite vs. Aorta groups, P = .76). There were also no significant differences in the IVUS data for the LITA grafts between the 2 groups (Table 7).

## Data from quantitative coronary angiography

QCA analysis was performed for all 50 patients. The average lengths of the SV and LITA segments analyzed by the QCA were 70.5 ± 18.8 mm and 84.0 ± 9.4 mm, respectively, on the early postoperative angiogram, and 61.3 ± 19.4 mm and 76.9 ± 14.2 mm, respectively, on the 1-year angiogram. The results of the QCA evaluation of the SV grafts at the 1-year angiogram were consistent with the IVUS evaluation. There was no difference in luminal diameter between the 2 groups (3.06 ± 0.69 vs. 3.09 ± 0.54 mm in the Composite and Aorta groups, P = .86), and the changes in luminal diameter from early angiograms to 1-year angiograms were not significantly different between the groups (P = .29). However, the QCA of the LITA grafts demonstrated marginally significant changes in luminal diameter from early angiograms to 1-year angiograms between the groups (P = .09) despite comparable initial luminal diameters on early angiograms (Table 8, Fig 5).

**Table 6. Comparison of IVUS measurements for SV grafts 1 year after CABG between the 2 groups.**

| Variables | Total (n = 46) | Composite group (n = 22) | Aorta group (n = 24) | P |
|---|---|---|---|---|
| Vessel area, mm² | 13.92 ± 4.94 | 14.13 ± 6.04 | 13.73 ± 3.80 | >.99 |
| Luminal area, mm² | 10.87 ± 4.14 | 11.05 ± 5.19 | 10.70 ± 2.99 | .97 |
| Intima-media area, mm² | 3.06 ± 1.02 | 3.08 ± 1.03 | 3.03 ± 1.03 | .90 |
| Proportion of intima-media area to vessel area, % | 22.6 ± 4.5 | 23.2 ± 4.7 | 22.0 ± 4.2 | .38 |
| Vessel diameter, mm | 4.15 ± 0.74 | 4.15 ± 0.91 | 4.15 ± 0.56 | >.99 |
| Luminal diameter, mm | 3.65 ± 0.70 | 3.65 ± 0.89 | 3.66 ± 0.49 | .97 |
| Intima-media thickness, mm | 0.25 ± 0.05 | 0.25 ± 0.04 | 0.24 ± 0.06 | .99 |
| Ratio of Intima-media thickness to vessel diameter, % | 12.0 ± 2.6 | 12.4 ± 2.7 | 11.7 ± 2.4 | .38 |

Continuous variables are presented as the means ± standard deviation.

CABG, coronary artery bypass grafting; IVUS, intravascular ultrasound; SV, saphenous vein.

**Table 7. Comparison of IVUS measurements for LITA grafts 1 year after CABG between the 2 groups.**

| Variables | Total (n = 45) | Composite group (n = 21) | Aorta group (n = 24) | P |
|---|---|---|---|---|
| Vessel area, mm² | 7.80 ± 1.90 | 8.26 ± 2.06 | 7.40 ± 1.70 | .13 |
| Luminal area, mm² | 5.83 ± 1.65 | 6.32 ± 1.75 | 5.41 ± 1.46 | .06 |
| Intima-media area, mm² | 1.97 ± 0.55 | 1.94 ± 0.49 | 1.99 ± 0.61 | .76 |
| Proportion of intima-media area to vessel area, % | 25.8 ± 6.0 | 24.0 ± 4.8 | 27.3 ± 6.6 | .07 |
| Vessel diameter, mm | 3.13 ± 0.40 | 3.22 ± 0.42 | 3.05 ± 0.37 | .16 |
| Luminal diameter, mm | 2.70 ± 0.39 | 2.81 ± 0.41 | 2.60 ± 0.36 | .08 |
| Intima-media thickness, mm | 0.22 ± 0.05 | 0.20 ± 0.04 | 0.22 ± 0.06 | .26 |
| Ratio of Intima-media thickness to vessel diameter, % | 13.9 ± 3.5 | 12.9 ± 2.8 | 14.8 ± 3.9 | .06 |

Continuous variables are presented as the means ± standard deviation.

CABG, coronary artery bypass grafting; IVUS, intravascular ultrasound; LITA, left internal thoracic artery.

## Histologic evaluation of the saphenous vein

The SV specimens were evaluated in all 50 patients. There was no significant difference in the baseline intima-media thickness between the 2 groups (546 ± 105 μm and 536 ± 122 μm in the Composite and Aorta groups, P = .77).

## Discussion

The present study demonstrated no significant difference in the intima-media thickness of the NT-SV graft evaluated by IVUS 1 year after surgery between the Y-composite and aortocoronary configurations, and no patient had abnormal plaque formation on IVUS.

A previous RCT, the SAVERITA trial, compared the RITA and SV as a second graft with the Y-composite configuration and showed that SV grafts demonstrated noninferior 1-year and 5-year patency rates compared with RITA grafts [7]. The SAVERITA trial suggested the following theoretical advantages of the SV as a Y-composite graft: (1) exposure to less circulatory stress in the Y-composite graft than the aortocoronary graft; and (2) the beneficial effect of endothelial-protective substances, such as NO, that are produced by the LITA and flow down to the composite SV graft [16,17]. However, inspired by previous studies showing excellent outcomes of NT-SV grafts used as an aortocoronary configuration [6,18], the present study was conducted to compare changes in the NT-SV used as a Y-composite and an aortocoronary conduit.

**Table 8. Comparison of QCA measurements for LITA and SV grafts at early and 1-year angiographic evaluation after CABG between the 2 groups.**

| Variables | Total (n = 50) | Composite group (n = 25) | Aorta group (n = 25) | P |
|---|---|---|---|---|
| Early | | | | |
| LITA diameter, mm | 2.15 ± 0.30 | 2.20 ± 0.29 | 2.10 ± 0.32 | .26 |
| SV diameter, mm | 4.03 ± 0.77 | 4.22 ± 0.87 | 3.84 ± 0.61 | .08 |
| 1 year | | | | |
| LITA diameter, mm | 2.09 ± 0.37 | 2.18 ± 0.39 | 2.00 ± 0.34 | .08 |
| SV diameter, mm | 3.08 ± 0.62 | 3.06 ± 0.69 | 3.09 ± 0.54 | .86 |

Continuous variables are presented as mean ± standard deviation.

CABG, coronary artery bypass grafting; LITA, left internal thoracic artery; QCA, quantitative coronary angiography; SV, saphenous vein.

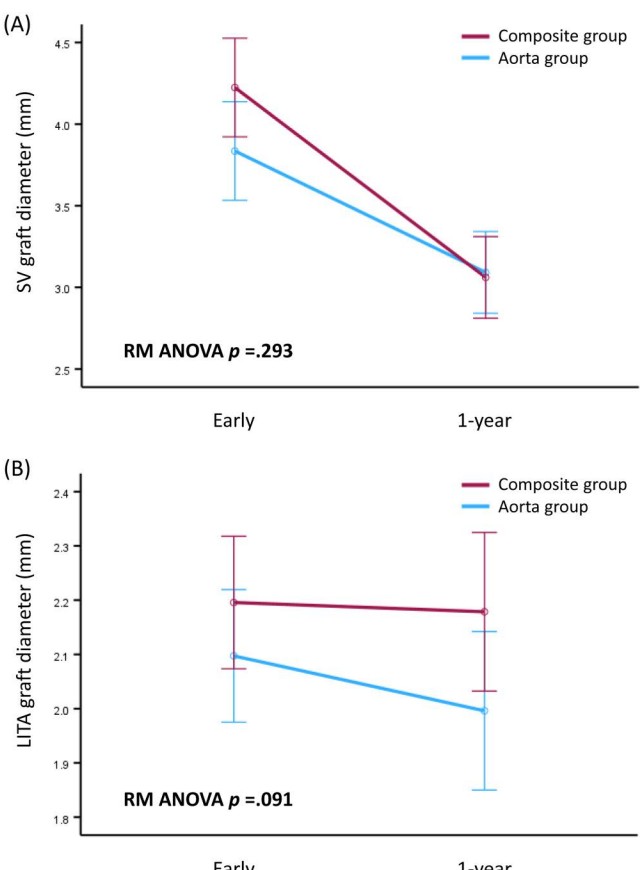

**Fig 5. Changes in luminal diameter from early angiograms to 1-year angiograms (A) in SV grafts and (B) in LITA grafts analyzed by QCA.**

The results of the present study suggest that NT-SV may be beneficial for both Y-composite and aortocoronary grafting. The NT technique has many beneficial effects on the patency of SV grafts. Endothelial integrity is preserved in the NT technique via the avoidance of high-pressure dilation to prevent spasm [19]. The vasa vasorum, which comprises a microvessel network of the vascular wall, is also preserved, and ischemia of the SV graft endothelium is prevented [20]. Recent evidence has provided great enthusiasm for the beneficial effect of the perivascular adipose tissue (PVAT)

surrounding the NT-SV on improving patency. The PVAT and intact adventitia present a "natural external stent" that supports the vein and mitigates the negative effect of pulsatile stress, which reduces the intimal hyperplasia and atherosclerosis, and potentially improves graft patency [21]. The PVAT surrounding the NT-SV also provides mechanical support that prevents longer grafts from kinking, which decreases early graft failure [21,22]. PVAT is a source of NO, adipocyte-derived factors, leptin, and adiponectin, which may play roles in reducing spasm, inhibiting atherosclerosis and thrombosis, and regulating of vascular tone [22–25]. Another study suggested that PVAT surrounding the SV had less metabolic dysfunction-driven chronic inflammation or "metaflammation" and consequent adipose tissue remodeling, including fibrosis, than the PVAT surrounding the coronary artery [26]. These beneficial effects may affect the favorable results of NT-SV grafts even when they are used in aortocoronary configurations. A major concern of harvesting SV with the surrounding PVAT might be an increased risk of leg wound complication. Although it was beyond the scope of the present study, this caveat could be solved by various efforts including topical application of autologous platelet rich plasma [27].

The grafting strategy of the Y-composite configuration based on in situ LITA has obvious benefits that the grafts can be used efficiently with shorter harvested lengths, and manipulation of the atherosclerotic aorta can be avoided. In contrast, competitive flow may occur and compromise the long-term patency of the graft [28–30]. The advantages of the aortocoronary SV graft include that the graft exhibits high and sufficient blood flow to target vessels immediately after anastomosis, which are not affected by competitive flow [29]. Exposure to the high diastolic pressure and sheer force of the ascending aorta were suggested as the theoretical drawbacks of aortocoronary SV grafts, but the clinical implications are not clear [30].

In conclusion, CABG using the NT-SV resulted in favorable 1-year patency without any abnormal changes in the lumen when it was used for both Y-composite and aortocoronary configurations. Longer-term follow-up in a larger population may be required to clarify the findings of the present trial.

## Limitations

The present study has several limitations. First, although this study was designed as a randomized controlled trial and power calculations were strictly performed, the sample size was relatively small. Second, this study included only patients who underwent off-pump CABG, which limits its generalizability to all CABG patients. Third, information on IVUS at baseline was absent, although we tried to supplement it with microscopic findings from the SV specimens.

## Conclusions

The intima-media thickness of the Y-composite NT-SV grafts was not significantly different from the aortocoronary NT-SV grafts 1 year after CABG.

## Supporting information

**S1 Fig. Microscopic examination of the saphenous vein grafts.**
(DOCX)

**S1 Table. Comparison of early postoperative angiographic patency rates between the 2 groups.**
(DOCX)

**S1 Data. Data.**
(XLSX)

**S1 Study Protocol. Study protocol.**
(PDF)

**S1 CONSRT checklist. CONSORT-2010-checklist.**
(DOC)

## Acknowlegments

The authors thank the Medical Research Collaborating Center, Seoul National University Hospital, for the statistical analyses and consultation.

## Author contributions

**Conceptualization:** Jeehoon Kang, Ho Young Hwang.

**Data curation:** Suk Ho Sohn, Yoonjin Kang, Ji Seong Kim, Jeehoon Kang, Ho Young Hwang.

**Formal analysis:** Suk Ho Sohn.

**Investigation:** Suk Ho Sohn, Jeehoon Kang, Ho Young Hwang.

**Supervision:** Ho Young Hwang.

**Validation:** Jeehoon Kang, Ho Young Hwang.

**Writing – original draft:** Suk Ho Sohn.

**Writing – review & editing:** Jeehoon Kang, Ho Young Hwang.

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
