## [Decision Letter · Decision Letter 0]

23 Jan 2025

PONE-D-24-50175

Morphologic Changes of the No-Touch Saphenous Vein as Y-composite Versus Aortocoronary Grafts (CONFIG Trial)

PLOS ONE

Dear Dr. Hwang,

Thank you for submitting your manuscript to PLOS ONE. After careful consideration, we feel that it has merit but does not fully meet PLOS ONE’s publication criteria as it currently stands. Therefore, we invite you to submit a revised version of the manuscript that addresses the points raised during the review process.

**ACADEMIC EDITOR: ** After a critical external peer review by two experts, I recommended a revision to improve the paper's clarity and presentation based on the reviewers' concerns. Please see the attached reviewer comments below.

We look forward to receiving your revised manuscript.

Kind regards,

Dr Redoy Ranjan, MBBS, MRCSEd, Ch.M., MS (CV&TS), FACS

Academic Editor

PLOS ONE

Journal Requirements:

Reviewers' comments:

Reviewer's Responses to Questions

**Comments to the Author**

1. Is the manuscript technically sound, and do the data support the conclusions?

Reviewer #1: Yes

Reviewer #2: Yes

2. Has the statistical analysis been performed appropriately and rigorously? 

Reviewer #1: Yes

Reviewer #2: Yes

3. Have the authors made all data underlying the findings in their manuscript fully available?

Reviewer #1: Yes

Reviewer #2: Yes

4. Is the manuscript presented in an intelligible fashion and written in standard English?

Reviewer #1: Yes

Reviewer #2: Yes

5. Review Comments to the Author

Reviewer #1: As the statistical reviewer I will focus on methods and reporting. The paper is well written, clear and with appropriate methods. The power calculations are also appropriate and there is broad balance between the groups. I only have a minor point to raise.

1) Can the authors clarify in the methods section how they planned to deal with missing data? were all the data complete? if yes, please state so. if not, why wasn't multiple imputation or another appropriate approach considered.

Reviewer #2: This paper aims to compare 1-year morphologic changes of the no-touch saphenous vein graft as Y-composite(Composite group) versus aortocoronary(Aorta group) configurations in coronary artery bypass grafting. The paper is well structured in all parts. I suggest to revise the language and to add the following paper in the discussion: 10.31083/j.fbe1402012

6. PLOS authors have the option to publish the peer review history of their article (what does this mean? ). If published, this will include your full peer review and any attached files.

**Do you want your identity to be public for this peer review?** For information about this choice, including consent withdrawal, please see our Privacy Policy .

Reviewer #1: No

Reviewer #2: No

---

## [Author Response · Author response to Decision Letter 1]

17 Feb 2025

Re: PONE-D-24-50175

Morphologic Changes of the No-Touch Saphenous Vein as Y-composite Versus Aortocoronary Grafts (CONFIG Trial)

Dear Editors and Reviewers,

We have revised our manuscript entitled “Morphologic Changes of the No-Touch Saphenous Vein as Y-composite Versus Aortocoronary Grafts (CONFIG Trial)”. Followings are the point-by-point responses to the reviewers’ comments and the changes we have made;

Reviewer #1:

Comment 1

As the statistical reviewer I will focus on methods and reporting. The paper is well written, clear and with appropriate methods. The power calculations are also appropriate and there is broad balance between the groups. I only have a minor point to raise.

1) Can the authors clarify in the methods section how they planned to deal with missing data? were all the data complete? if yes, please state so. if not, why wasn't multiple imputation or another appropriate approach

Response 1

There were 1 missing data out of 25 patients in the Composite group and 3 missing data out of 25 patients in the Aorta group in terms of primary endpoint.

As Reviewer indicated, we additionally performed multiple imputation to deal with missing data, and revised the manuscript accordingly.

Change 1

Line 176

For the primary endpoint, multiple imputation was additionally performed to deal with missing data using Fully Conditional Specification (FCS) method.

Line 224

When further analysis was performed using multiple imputation to deal with missing data, the intima-media thicknesses of the SV grafts also demonstrated no difference between the groups (0.25 ± 0.06 mm vs. 0.24 ± 0.06 mm in Composite vs. Aorta groups, P =.76).

Reviewer #2:

Comment 1

This paper aims to compare 1-year morphologic changes of the no-touch saphenous vein graft as Y-composite(Composite group) versus aortocoronary(Aorta group) configurations in coronary artery bypass grafting. The paper is well structured in all parts. I suggest to revise the language and to add the following paper in the discussion: 10.31083/j.fbe1402012

Response 1

As Reviewer indicated, we cited the article by Jiritano et al in the manuscript.

Change 1

Line 271

A major concern of harvesting SV with the surrounding PVAT might be an increased risk of leg wound complication. Although it was beyond the scope of the present study, this caveat could be solved by various efforts including topical application of autologous platelet rich plasma [27].

In addition, references were renumbered as they appeared in the text.

Sincerely yours,

Ho Young Hwang

Professor

Department of Thoracic & Cardiovascular Surgery

Seoul National University Hospital

Seoul, Korea

---

## [Decision Letter · Decision Letter 1]

19 Mar 2025

Morphologic Changes of the No-Touch Saphenous Vein as Y-composite Versus Aortocoronary Grafts (CONFIG Trial)

PONE-D-24-50175R1

Dear Dr. Hwang,

We’re pleased to inform you that your manuscript has been judged scientifically suitable for publication and will be formally accepted for publication once it meets all outstanding technical requirements.

Kind regards,

Dr Redoy Ranjan, MBBS, MRCSEd, Ch.M., MS (CV&TS), FACS

Academic Editor

PLOS ONE

Review Comments to the Author

Reviewer #1: The authors have done a good job in adressing my points previously raised and I have nothing further to add.

Reviewer #2: This randomized controlled trial was aimed to compare 1-year morphologic changes of the no-touch saphenous vein graft as Y-composite(Composite group) versus aortocoronary(Aorta group) configurations in coronary artery bypass grafting. The paper is well structured but I suggest to revise the language

---

## [Editor Report · Acceptance letter]

PONE-D-24-50175R1

PLOS ONE

Dear Dr. Hwang,

I'm pleased to inform you that your manuscript has been deemed suitable for publication in PLOS ONE. Congratulations! Your manuscript is now being handed over to our production team.

Kind regards,

on behalf of

Dr. Redoy Ranjan

Academic Editor

PLOS ONE